# Emotional Responses and Support Needs of Healthcare Professionals after Adverse or Traumatic Experiences in Healthcare—Evidence from Seminars on Peer Support

**DOI:** 10.3390/ijerph20095749

**Published:** 2023-05-08

**Authors:** Katja Schrøder, Elisabeth Assing Hvidt

**Affiliations:** Department of Public Health, University of Southern Denmark, 5000 Odense, Denmark

**Keywords:** health worker safety, patient safety, psychological support, peer support, second victim

## Abstract

The aim of this study was to identify (i) emotions experienced by healthcare professionals (HCPs) after adverse or traumatic events and (ii) needs for support after adverse or traumatic events. Data for this qualitative, descriptive study were collected at 27 seminars for 198 HCPs introducing a peer-support programme after adverse or traumatic events (The Buddy Study). Through interactive exercises, participants shared their experiences, and this study reports on the responses of an exercise identifying emotions and needs after an adverse or traumatic event. The top five emotions were anger, guilt, impotence, grief, and frustration and anxiety, and the top five needs were to be met with understanding, recognition, listening, care, and respect. Ten categories of emotions experienced by HCPs after adverse or traumatic events were constructed, and the five categories with the highest number of mentions were anger and impotence, fear and insecurity, negative self-evaluation, guilt and shame, and alone and overloaded. Nine categories relating to needs for support after adverse or traumatic events were constructed, and the five categories with the highest number of mentions were: being seen and understood, compassion, being respected, time to recover, and organisational support. The emotional disclosure promoted at the peer seminars of the Buddy Study revealed that all participants share the same emotional distress, being either second victims or potential second victims. Moreover, the support needed was of a human-to-human nature that all participants felt capable of providing as a “buddy” for a colleague. Both the identified emotions and needs for support identified in this study may contribute to qualifying the development of the content of support programmes for HCPs after traumatic or adverse events.

## 1. Introduction

Adverse events and unexpected outcomes in healthcare impact not only patients and relatives but also the involved healthcare professionals (HCPs), often named ‘second victims’ in the literature [1,2,3]. Second victims are defined as healthcare providers who are involved in an unanticipated adverse patient event, in a medical error and/or a patient-related injury, and become victimized in the sense that the provider is traumatized by the event [2]. Second victims report a high prevalence and wide range of physical and psychological symptoms, such as troubling memories, anxiety, anger, remorse, guilt, distress [3,4], hypervigilance, and doubts about knowledge and skill [5]. The higher the degree of patient harm, the longer the second victim symptoms seem to last [5]. Second victim support is critical for psychosocial and physical recovery after an adverse event [6]. This is paramount not only for the individual HCP but also for the patients, since the health and wellbeing of the staff are crucial for the quality and safety provided in healthcare [4,6,7,8,9].

Through a literature review, Seys et al. [7] identified considerations and interventional strategies to support second victims. Suggested considerations are that the time between an adverse event and support is crucial, that structured sessions need to be provided, that highly respected physicians or physicians in a senior position are encouraged to discuss their errors and feelings, that programmes which are focused on preventing, identifying, and treating burnout are developed, and, finally, that empathy within the team is promoted. They propose the following strategies: talk and listen to second victims, organize and facilitate open discussion of the error, share experiences with peers, organize special conferences on the issue of second victims to increase awareness, and provide a professional and confidential forum to discuss their errors and inquire about colleague coping. In a systematic review, Seys et al. [9] found only 12 second-victim support programmes implemented between 2006 and 2017, and the authors recommend that support for HCPs after adverse events receives further attention and priority.

One of the earliest organisations to implement a second victim intervention was the Medically Induced Trauma Support Services (MITSS) in 2002, which was developed in partnership by a patient and an anaesthesiologist involved in an unanticipated event that seriously harmed the patient [10]. At the University of Missouri Health Care (MUHC), the forYOU programme was implemented in 2009. This is an evidence-based emotional support structure for second victims based on research with recovering second victims. Members of the forYOU Team provide emotional support using The Scott Three-Tiered Interventional Model of Second Victim Support. The model consists of three tiers with the nature of support escalating from Tier 1 (local support at the unit or department), through Tier 2 (trained peer supporters) to Tier 3 (referral network with access to professional support). Each tier provides increasing institutional resources to help ensure that the emotional needs of the clinician are met [2,10]. Through a collaboration with MUHC researchers, the Nationwide Children’s Hospital replicated the forYOU programme in a local context (YOU Matter) in 2013 [11]. Other institutions have adapted elements from the forYou programme when developing second victim support, e.g., the Resilience In Stressful Events (RISE) programme at Johns Hopkins Hospital [12], the Healing Emotional Lives of Peers (HELP) programme at Mayo Clinic [13], and the Second Victim Support at the Clinico San Cecilio University Hospital in Granada, Spain [14]. 

The ERNST (European Researchers’ Network Working on Second Victims) Consortium identified current lines of study in Europe on the phenomenon of second victims to provide an approximate overview of second victim research and support programmes. They identified seventeen ongoing interventions to support second victims [15]. In Denmark, we have developed, implemented, and evaluated a peer-support programme, The Buddy Study, in two departments at Odense University Hospital. A full description of the programme is discussed in the evaluation study [16]. 

Evidently, second victim support programmes are emerging, but only a few studies have provided preliminary data on the beneficial effects of the support programmes [9]. This is still a relatively new research field, and the development, implementation, and evaluation of support programmes require several years. During this process, a continuing exploration of the needs of second victims is required to design and test different types of interventions. Several studies have found that the most desired form of support after involvement in traumatic events is talking to peers [2,13,17,18,19]. However, ‘talking to peers’ is a relatively undefined term with little concrete assignments on what and whom these talks involve. 

The purpose of this study is to contribute to the sparse body of evidence on the support needs of second victims. The objective was to identify (i) emotions experienced by HCPs after adverse or traumatic events and (ii) needs for support after adverse or traumatic events. 

## 2. Materials and Methods

This is a qualitative descriptive study.

### 2.1. Participants and Setting

From May 2018 to April 2019, the two authors developed and conducted teaching at 27 seminars for a total of 198 participants. The seminars were a compulsory part of the abovementioned Buddy Study—a peer-support programme for HCPs after adverse events [16]. HCPs from two departments at Odense University Hospital in Denmark participated in the Buddy Study program: midwives in the Department of Obstetrics and Gynaecology in Odense and Svendborg and physicians at the Internal Medicine and Emergency Department in Svendborg. Both departments are involved in high degrees of acute patient care and decision making. 

### 2.2. The Buddy Study Program

The objective of the Buddy Study programme was to facilitate peer support after adverse or traumatic events through a formalised buddy system. The programme encompassed a compulsory seminar about second victims and peer support for two designated groups of HCPs, self-selection of two buddies, and a system for buddy activation and response after adverse or traumatic events. During the seminars, adverse events were defined as patient events with unanticipated adverse outcomes, medical errors, and/or patient-related injuries. Traumatic events could include situations not associated with safety incidents, such as patient death or workplace violence. The buddy system could be activated through the involved HCP, a colleague, or the manager. The buddy encounters should happen outside working hours to ensure a space for private, unguarded conversations, and the buddy would be paid for 2 h of work. 

The Buddy Study programme was founded on five underlying principles: (i) Recognition of exposure to adverse or traumatic events as a fundamental condition for HCPs. (ii) Organisational responsibility towards all employees every time and not based on inconsistent or random assessments of when support or debriefings should be facilitated. (iii) Relationships are of central importance, and HCPs should be able to select a peer supporter of their own choice. (iv) Build on existing resources in the departments by involving the HCPs who are already trained to care for people in crisis. (v) Research-based evaluation of the intervention. 

The purpose of the two-hour seminars was to give participants knowledge about the second victim phenomenon and the physical and emotional responses that may be associated with a traumatic or adverse event. The seminars were conducted for smaller groups of 5–15 participants at the time, and they addressed all participants as both potential second victims and peer supporters (buddies). The content included presentations of cases and theories, smaller exercises in pairs, and plenary exercises to create mutual awareness of support needs after a traumatic or adverse event. 

### 2.3. Data Collection

This study reports on the responses of an exercise named *Inside experiences,* which is an adaption of an exercise developed as a model for teaching HCPs about how to meet and support patients in existential crises [20]. The exercise was two-fold, and the first part was the presentation of the following question: *Think of an adverse event or a traumatic experience you have been involved in—which emotions or themes do you connect with this experience?* Without revealing any information about the events, participants voiced the words or expressions describing their emotions and reactions related to the event. All words were written on the left side of a white board. When no more words came up, the second part of the exercise was initiated with the following question: *Think of the same situation again. How would you have liked to be met, what were your needs?* Those words or expressions were written on the right side of the white board. 

### 2.4. Analysis

After each seminar, the white board was photographed, and all words and expressions were subsequently extracted to an excel spreadsheet in alphabetical order. Identical words were listed only once, but a separate column kept track of number of mentions. This quantification of words is in accordance with the first step of summative content analysis [21]. Following this, a process of interpretation of content was carried out through clustering and categorisation of all words by the first author (KS). The second author (EAH) read this analysis and proposed alternative interpretations for some categories. All words and developed categories were reviewed and discussed by both authors, and the final categorisation was developed. For transparency of the construction and content of each category, all words are listed in Table 1 and Table 2. To ensure accuracy, the translation process from Danish to English was conducted by the authors in collaboration with linguists at the language service unit at University of Southern Denmark. 

## 3. Results

Accumulated for the 27 seminars, a total of 584 words or expressions were logged. 

The first part of the exercise (Think of an adverse event or a traumatic experience you have been involved in—which emotions or themes do you connect with this experience?) generated a total of 307 words or expressions. Of these, 49 were mentioned more than once (2–19 times), and 56 were mentioned once. The top five emotions were: anger (19 mentions), guilt (18 mentions), impotence (14 mentions), grief (13 mentions), and frustration and anxiety (12 mentions respectively).

The second part of the exercise (*Think of the same situation again. How would you have liked to be met, what were your needs?*) generated a total of 277 words or expressions, of which 45 were mentioned more than once (2–23 times), and 42 were mentioned once. The top five needs were to be met with: understanding (23 mentions), recognition (21 mentions), listening (15 mentions), care (14 mentions), and respect (12 mentions). 

All words and expressions are displayed in Table 1 and Table 2.

We constructed ten categories of emotions experienced by HCPs after adverse or traumatic events: (i) anger and impotence, (ii) fear and insecurity, (iii) negative self-evaluation, (iv) guilt and shame, (v) alone and overloaded, (vi) grief and sorrow, (vii) physical manifestations, (viii) positive self-evaluation, (ix) collaboration and communication, and (x) existential thought. We constructed nine categories of needs for support after adverse or traumatic events: (i) being seen and understood, (ii) compassion, (iii) being respected, (iv) time to recover, (v) organisational support, (vi) professional support and supervision, (vii) existential needs, (viii) being a part of a team, and (ix) positive sense of self. 

For a clear overview, Table 3 shows the constructed categories and the total number of mentioned words or expressions within each category. The categories are listed descending from the most mentioned to the least mentioned. 

## 4. Discussion

The words and expressions listed in part one of the *Inside experiences* exercise have many similarities with previous findings on psychological and psychosomatic symptoms among second victims [3,4,5,6].

Although research articles on the impact of adverse and unexpected outcomes in healthcare have increased dramatically in recent years [4], the literature appears to be out of reach for most HCPs; thus, the individual remains alone in their personal experience [22]. Austin et al. (2022) found a socially constructed pattern of behaviour of not talking about the emotional impact of adverse events in an action research study developing and evaluating a support tool. The silence was perpetuated since HCPs continued to behave in a way they thought was expected (hiding emotional response), and, furthermore, this perceived expectation inhibited providing supportive actions towards colleagues [22]. Consequently, there seems to be a need for educational measures in local settings addressing the cultural barriers to disclose and normalise emotional responses to adverse events. This was incorporated in our peer-support programme, the Buddy Study, where one of five underlying principles was *Recognition of exposure to adverse or traumatic events as a fundamental condition for HCPs* [16]. This principle was communicated at the seminars along with knowledge about the second victim phenomenon to facilitate the understanding that adverse or traumatic events are inevitable in healthcare, and collegial support and compassion are required in the aftermath of these events. The seminars received tremendously positive evaluations, and 98% agreed that they gained knowledge about the second victim phenomenon; 93% believed that compulsory participation in the seminar provided mutual insight and understanding [16]. 

Part one of the *Inside experiences* exercise enabled all participants to realize that their colleagues had been affected by similar emotional responses as themselves. It seemed to resonate immensely with everyone in the room to see and recognize the emotions as they were listed on the white board. Correspondingly, Austin et al. [22] found that intensifying the visibility of the impact of adverse or critical events enabled a realisation of the normality of such experiences.

Part two of the *Inside experiences* exercise brings some novel findings about the support needs of HCPs after adverse events. Several studies found that the most desired form of support after involvement in traumatic events is talking to peers [2,13,17,18,19], but this study provides insights on how HCPs would like to be met in these peer-to-peer talks. Concerns about whether buddies in this programme are adequately trained for the task could be raised. Austin et al. described a consensus between HCPs that “none of us know how to deal with someone who is distressed or upset” despite them working in mental health and supporting clients in their distress [p. 6]. However, part two of the *Inside experiences* exercise made the participants of this study aware of the exact opposite: the needs expressed were needs that all participants felt capable of meeting as a buddy for someone else. Providing a relational space for compassionate, respectful listening, where the person feels *seen* and *understood,* and offering professional support and supervision on the medical treatment and care provided were traits the participants seemed confident to deliver when this was discussed after part two of the exercise. This is reflected in the evaluation of the seminar, where 88% agreed that they felt prepared to become a buddy for their colleague [16], a finding that may contribute to the understanding that HCPs *do know* how to deal with colleagues in emotional distress if they are provided with knowledge about second victims’ reactions and needs. 

These findings seem to be opposing the consensus in other support programmes, where additional training, attendance at meetings, simulation exercises, and/or post-encounter debriefings for all peer responders are required [12,23]. In this programme, we considered all HCPs to be generally qualified to provide support for human beings in crisis, since this is the trust society puts in them when citizens are taken ill or encounter trauma or death as patients or relatives. This is also the core competence for HCPs as buddies, providing psychological first aid for a colleague after an adverse event. One may even argue that many of the needs expressed by the participants call for skills or capacities that almost all human beings possess—the ability to listen, mentalize, and respond compassionately.

In an evaluation of the volunteer peer responders in the abovementioned Resilience In Stressful Events (RISE) programme, the responders found their duties to be meaningful, personally satisfying, and positively impactful [24]. The study also found that being a peer responder contributed to professional and personal growth, empowerment, and resilience. Similarly, in the Buddy Study, responders (or ‘buddies’) mainly had positive experiences of being able to help their colleague and found that it had been an opportunity to reflect upon their own experiences with adverse events [16]. Accordingly, one may consider the beneficial effects not only for the second victims but also for the responders in peer-support programmes. However, we suggest long-term monitoring of the responders’ experiences of added workload, psychological distress, and burnout when implementing peer-support systems. 

Since many support programmes are tailored from The Scott Three-Tiered Interventional Model of Second Victim Support [2], one may reflect on the trade-off between self-selected relations among the entire group of colleagues (within the same department) and specific training of a selected group of peers (hospital-wide support team). A Dutch study reporting on interviews with physicians and quality and safety staff members found that a hospital-wide solution for peer support was not requested, since it was considered to emphasise the existing cultural problem with low acceptance of vulnerability and support needs [19]. The participants felt that talking to a peer with the same background and training would provide more qualified professional assessment of clinical decision making than calling a hotline from a hospital-wide support team.

*Relationships are of central importance* was yet another of the underlying principles of the Buddy Study, where HCPs selected peer supporters (buddies) of their own choice, i.e., already established and safe relations with colleagues in the same department. This approach differs from the institutional support systems, and this was positively emphasised during the interviews in the evaluation of the study: self-selected relations were considered to add a greater sense of safety and to encourage a general sense of responsibility towards each other [16]. These findings correspond well with the emotions experienced by HCPs after adverse or traumatic events in this study. Emotions, such as anger, impotence, fear, insecurity, inadequacy, self-blame, guilt, shame, and loneliness, may be kept in a personal space and only shared in safe, trusted relations. According to Robert D. Stolorow’s trauma theory, providing a validating and responsive environment in which painful emotions can be listened to, understood, and contained, what he calls “a relational home”, results in a healing process [25]. Stolorow emphasizes the therapeutic effect of being understood by someone who shares the same experience horizon, what he calls “a sister or brother who knows the same darkness” [p. 49], since this paves the way for a deep-felt emotional support motivated by feelings of solidarity and empathy. 

### Strengths and limitations

This study provides a valuable and transparent picture of the emotions experienced by HCPs after adverse or traumatic events and their needs for support. However, since the analysis is based on data consisting of single words or expressions, it lacks in-depth descriptions and contextualization of the felt emotions of the HCPs’ experiences with adverse or traumatic events. Accordingly, we found the analytical process remarkably challenging because the clustering of single words and expressions into categories seemed to hold endless opportunities for interpretations. Full transparency of all words and expressions is provided in Table 1 and Table 2. However, the summative and, thus, quantifying approach to the subject at hand utilized in this study provides a unique opportunity to capture the volume and density of HCPs’ emotional distress as well as support needs in relation to adverse or traumatic events. 

## 5. Conclusions

The aim of the study was to identify (i) emotions experienced by HCPs after adverse or traumatic events and (ii) needs for support from their surroundings after adverse or traumatic events. The data were collected at seminars (*n* = 27) introducing a peer-support programme for HCPs after adverse and traumatic events (The Buddy Study), where participants (*n* = 198) expressed emotions and needs in single words or expressions. Summative content analysis demonstrated that the top five emotions were *anger, guilt*, *impotence*, *grief*, and *frustration and anxiety*, and the top five needs were to be met with *understanding*, *recognition*, *listening*, *care*, and *respect*. Through clustering and categorisation of all words, ten categories of emotions experienced by HCPs after adverse or traumatic events were constructed. The five categories with the highest number of mentions were *anger and impotence*, *fear and insecurity*, *negative self-evaluation*, *guilt and shame*, and *alone and overloaded*. Nine categories of needs for support after adverse or traumatic events were constructed, and the five with the highest number of mentions were: *being seen and understood*, *compassion*, *being respected*, *time to recover*, and *organisational support*. 

All participants shared the same emotional distress, being either second victims or potential second victims. Moreover, the support needed was of a human-to-human nature that all participants felt capable of providing as a “buddy” for a colleague. This is important knowledge for managers and leaders of support programmes, since the identified needs and emotions in this study may contribute to qualifying the development of the content of second victim support programmes in other departments or organisations. 

## Figures and Tables

**Table 1 ijerph-20-05749-t001:** Categorisation of words and numbers of mentions (*n* = 307) from seminar exercise I: ‘Think of an adverse event or a traumatic experience you have been involved in—which emotions or themes do you connect with this experience?’.

Anger and Impotence (70 Mentions)	Fear and Insecurity (55 Mentions)	Negative Self-Evaluation (35 Mentions)	Guilt and Shame (30 Mentions)	Alone and Overloaded (30 Mentions)	Grief and Sorrow (29 Mentions)	Physical Manifestations (17 Mentions)	Positive Self-Evaluation (16 Mentions)	Collaboration and Communication (13 Mentions)	Existential Thoughts (12 Mentions)
Anger (19) Impotence (14) Frustration (12) Feeling of powerlessness (10) Experience of injustice (5) Irritation (4) Disappointment (3) Unfairness (2) “Why me?” (1)	Anxiety (12) Fear (8) Doubt (8) Uncertainty (6) Panic (4) Insecurity (4) Shock (2) Doubting own competences (2) Fear of repetition (2) Afraid (1) Fear of having overlooked something (1) Fear of own inadequacy (1) Fear of complaint (1) Terror (1) Insecurity caused by surroundings (1)	Inadequacy (11) Self-blame (5) Incompetence (4) Self-scrutiny (3) Stupid (2) Ignorance (2) Adequate behavioural pattern? (1) Compromised professionalism (1) Lack of skills (1) Insufficient overview (1) Sloppy work (myself) (1) Inexperience (1) ‘I am a lousy doctor’ (1) Inferiority (1)	Guilt (18) Shame (7) Guilty conscience (2) Embarrassment (2) Rumination (1)	Aloneness (5) Vulnerability (5) Loneliness (3) Isolation (2) Afraid of what others may think (2) Emptiness (2) Thoughts going in circles (2) First and last thing on your mind (1) Attacked (1) Accused (1) Feeling judged (1) Everyday life is affected (1) Identity (1) Internal film (1) Let down (1) Racing thoughts (1)	Grief (13) Sorrow (9) Despair (3) Sadness (2) Unhappiness (1) Broken (1)	Fatigue (3) Lack of energy (2) Concentration difficulty (2) Rapid heartbeat (2) Loss of appetite Restlessness (physical) (1) Nausea (1) Gut punch (1) Tears (1) Hot flush (1) Hyperventilation (1) Uneasiness (1) Stomach ache (1)	Professional pride (4) Experience (2) Professionalism (1) Immersion (1) High level of competence and skills (1) Competences (1) Relief (to be able to act) (1) Overview (1) Being professional—setting myself aside (1) Being Professional—not becoming too involved (1) Content (1) Vigilant during the event (1)	Good teamwork (4) Poor communication (2) Disagreement (2) Poor collaboration (2) Discussion (1) Sense of community (1) Unprofessional behaviour (1)	Meaninglessness (4) Luck (1) Existential angst and pain (1) Basic terms when working in life/death situations (1) The frailty of life (1) Losing faith (in one’s God) (1) Coincidences (1) ‘Is it worth it?’ (1) Considering quitting/finding another job (1)

**Table 2 ijerph-20-05749-t002:** Categorisation of words and numbers of mentions (*n* = 277) from seminar exercise II: ‘Think of the same situation again. How would you have liked to be met, what were your needs?’.

Being Seen and Understood (88 Mentions)	Compassion (69 Mentions)	Being Respected (31 Mentions)	Time to Recover (27 Mentions)	Organisational Support (22 Mentions)	Professional Support and Supervision (14 Mentions)	Existential Needs (11 Mentions)	Being a Part of a Team (10 Mentions)	Positive Sense of Self (5 Mentions)
Understanding (23) Recognition, (21) Listening (15) Openness (10) Curiosity (5) Acceptance (3) To feel seen and heard (3) Attention/interest (3) Forthcomingness (2) Pat on the back, confirmation (2) To be left in peace if needed (1)	Care (14) Sense of security (10) Inclusiveness (9) Empathy (8) People extending their hands to you (5) Encouragement (3) Support (3) Compassion (2) Humaneness (2) A hug (2) Help to categorise the event (2) A hand to hold (1) Sympathy (1) Cautiousness (1) Physical closeness (1) Helpfulness (1) The courage to be present (1) Tolerance (1) Kindness (1) Warmth (1)	Respect (12) Non-judgemental (6) Trust (3) To be taken seriously (2) Equality (1) To tell my version of the story (1) That my colleagues know the true story (1) Appropriate communication in the context (1) Situational awareness (1) Avoid rumours or gossip (1) Apology—if someone has done you wrong (1) Respecting different emotional responses in the aftermath (1)	Time (11) Peace (7) Follow up (3) Talk through the course of the events (2) To be shielded from the situation (1) Clarification (1) Distance (1) Given space to recover (1)	Debriefing (6) Consideration in the planning of tasks on the following shifts (3) Supervision (2) Managerial support (2) That it is not neglected (1) Good communication (1) Organisational support (1) Constructive analysis (1) Prevention (1) Predictability (1) To be met instantaneously (1) Psychological counselling (1) Transparency in the handling of the aftermath (1)	Professional back-and-forth (3) Professionalism (3) Honest evaluation of the sequence of events (3) Constructive feedback (2) Professional evaluation of the sequence of events (1) Professional back-and-forth in an experienced team (1) Getting answers to why it turned out as it did (1)	Forgiveness (3) To put things into perspective with someone else (2) Acceptance (of the outcome) (2) To admit responsibility (1) To get closure (1) To find meaning in the event (1) To contain fallibility (as a human being) (1)	Community (3) Dialogue (2) To share the responsibility with the others on the shift (2) To be a part of a team (1) Going out for a drink/being together with colleagues (1) Practical help (1)	Strength (2) Gratitude (2) Humility—knowing that you did your best (1)

**Table 3 ijerph-20-05749-t003:** Constructed categories and the number of mentioned words or expressions within each category. Part one *n* = 307, part two *n* = 277.

Part One	Part Two
Think of an adverse event or a traumatic experience you have been involved in—which emotions or themes do you connect with this experience?	Think of the same situation again. How would you have liked to be met, what were your needs?
Anger and impotence (70 mentions)	Being seen and understood (88 mentions)
Fear and insecurity (55 mentions)	Compassion (69 mentions)
Negative self-evaluation (35 mentions)	Being respected (31 mentions)
Guilt and shame (30 mentions)	Time to recover (27 mentions)
Alone and overloaded (30 mentions)	Organisational support (22 mentions)
Grief and sorrow (29 mentions)	Professional support/supervision (14 mentions)
Physical manifestations (17 mentions)	Existential needs (11 mentions)
Positive self-evaluation (16 mentions)	Being a part of a team (10 mentions)
Collaboration and communication (13 mentions)	Positive sense of self (5 mentions)
Existential thoughts (12 mentions)	

## Data Availability

The data that support the findings of this study are available from the corresponding author upon reasonable request.

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
