# Peer review of "Emotional Responses and Support Needs of Healthcare Professionals after Adverse or Traumatic Experiences in Healthcare—Evidence from Seminars on Peer Support"

_ijerph, 2023, doi:10.3390/ijerph20095749_

Round 1

Reviewer 1 Report

The study addresses the experience of second victims through a qualitative analysis of the experiences of professionals participating in seminars focused on training in the experience of SV. Therefore, the participants already show a different interest in this topic than other colleagues.

Two key questions arise. The first, less novel, analyses emotions related to the SV experience. The second, preferences for emotional support a in the event of an adverse or traumatic event. This second question is new in the literature and is particularly pertinent.

A few comments that I hope will help the authors to some extent.

1. Summary, include the number of participants, not just the number of seminars.

2. Define SV and traumatic experience, how was it presented to the participants, what might be the differences in the framework of the Buddy programme? How participants were involved?

3. Differences in the experiences, preferences and comments between men and women

4. Describe the approach of these seminars - preventive, to be a supportive colleague, ... ?

5. Discussion, similarities and differences in the answers to the first question on the results of studies that have explored this issue previously, idem for the second question.

6. Utility of these results for managers, middle management, leaders of support programmes, lessons learned.

7. Suggestions for action, assessment of the extent to which they are realistic and can be put into practice and whether they could be generalised to most countries.

Reviewer 2 Report

Thank you for your work! This is such an important topic to address.

Say more about the choice of the “second victim” construct? Much of what is described in the paper, from the stressors, to the support offered could easily overlap with moral injury, moral distress, and/or burnout. Offering some discussion on why those constructs were not evaluated would be helpful (or even speak to them as potential variables to evaluate in future research of the intervention).

What role do health organizations and/or health policy play in the experience of the second victim construct by HCPs? It isn't only the exposure to difficult events in healthcare that may matter, but the organizational can greatly influence HCP experience.

The description of the Buddy study program was too brief, say a bit more about the details of the intervention and the theory behind it.

Speak more to limits of generalizability. Would reception of the program and/or responses differed in different clinical settings? Different health systems? What about selection bias?

Was there any control group or interviews with those that did not receive the intervention?

How was this project/study supported organizationally? Were workloads reduced for HCPs in order to give and receive peer support?

In some health systems, peer support may be negatively received because it is another demand on HCPs time when many systems are already short staffed and overloaded. Can you speak more to barriers/facilitators to implementing this intervention?

Articles that may be helpful in framing second victim relative to moral distress/injury:

Epstein, E. G., Whitehead, P. B., Prompahakul, C., Thacker, L. R., & Hamric, A. B. (2019). Enhancing understanding of moral distress: the measure of moral distress for health care professionals. AJOB empirical bioethics, 10(2), 113-124.

Riedel, P. L., Kreh, A., Kulcar, V., Lieber, A., & Juen, B. (2022). A scoping review of moral stressors, moral distress and moral injury in healthcare workers during COVID-19. International Journal of Environmental Research and Public Health, 19(3), 1666.

Ehman, A. C., Smith, A. J., Wright, H., Langenecker, S. A., Benight, C. C., Maguen, S., ... & Griffin, B. J. (2022). Exposure to potentially morally injurious events and mental health outcomes among frontline workers affected by the coronavirus pandemic. Psychological Trauma: Theory, Research, Practice, and Policy.

Reviewer 3 Report

Dear Authors,

Thank you for coming up with this relevant and interesting study. I would like you to consider the following:

1. Abstract - the Headings (Background, method, result, conclusion) maybe removed 

2. The design of the study is descriptive qualitative but the expression of the aims is more of a quantitative design and this was supported by the way the result was presented.

3.. Methods:

-Describe your involvement in the 27 seminars 

-Also clarify if the 198 participants attended all 27 seminars, did they attend altogether?

- any procedure for consent from the participants?

-Add a subheading for the intervention/instrument

-Did the participants answer individually or by group? Did you include all answers from the 198 participants? If not, what was the cut off point? 

-How did you determine if the participant's involvement in the traumatic event was as the main victim or the second victim? Your study focus is on the experience of the HCP as a second victim.

-Were the words and themes provided for them to choose from? Were there descriptions of the feelings? or just words with no further descriptions? Any notes on their emotions during the exercise? Any debriefing prepared for the activity?

Analysis- Was there a third (external) reviewer of the themes/categories developed? How did you control any source of bias?

Results:

I suggest that the tables 1 and 2 be organized and presented to be reader friendly. How do they differ with table 3?

The tables should be able to reflect the process of developing the themes and categories.

The results showed just list of the emotions and needs but no descriptions were provided. It would be nice to include some further verbalizations from the participants. The result seem just an enumeration of feelings and needs with no story behind the lists. The design is descriptive qualitative but with the presentation, it is more of a quantitative design by just counting the frequency of the words listed. Categorizing the listed words does not suffice for it to be a qualitative study. You have categorical data in a quantitative design. 

Discussion

- Where did the subheadings in the discussion come from? How do you connect the discussion with the results? These are not reflected in the presentation of your results.

Conclusion- It should not be a summary of the procedure and results of the study. From these findings, what now can you conclude?

Thank you and I look forward to the revised copy of your manuscript.

Round 2

Reviewer 3 Report

Dear authors,

Thank you for the revised manuscript. 

I would like to reiterate the addition of a subheading for the instrument/intervention  separate from the participants and setting under the methodology section. 

The definition and discussion of adverse and traumatic events should be in the introduction and not under participants and settings

Line 95-Objective of the study:  what do you mean by surrounding? I suggest that the process of identifying the emotions be articulated in the objectives.

Results: For the title of the tables, include the (n=?)  since you used frequency in summarizing and categorizing your data. 

Data analysis-  If during the exercise, the listed emotions and needs were summarized and processed before the participants, I suggest that this part should be mentioned as part of the validation of the data.

Thank you.
